Ontogeny of long-range vocalizations in a Neotropical fossorial rodent: the Anillaco Tuco-Tuco (Ctenomys sp.)

Amaya Juan Pablo juanentuculandia@gmail.com jamaya@crilar-conicet.gob.ar 1 2
Areta Juan Ignacio 3
1 Centro Regional de Investigaciones Científicas y Transferencia Tecnológica de La Rioja (CRILAR-CONICET) , Anillaco , La Rioja , Argentina
2 Universidad Nacional de La Rioja (UNLAR) , La Rioja , Argentina
3 Instituto de Bio y Geociencias del Noroeste Argentino (IBIGEO-CONICET) , Rosario de Lerma , Salta , Argentina
Zimmermann Elke
Electronic publication date: 2018 Feb 14
Publication date: 2018
Volume: 6
Electronic Location ID: e4334
Received 2017 Jun 27; Accepted 2018 Jan 17
Copyright: ©2018 Amaya and Areta
Copyright year: 2018
Copyright holder: Amaya and Areta
License: This is an open access article distributed under the terms of the Creative Commons Attribution License, which permits unrestricted use, distribution, reproduction and adaptation in any medium and for any purpose provided that it is properly attributed. For attribution, the original author(s), title, publication source (PeerJ) and either DOI or URL of the article must be cited.
License URL: https://creativecommons.org/licenses/by/4.0/

Keywords: Ctenomyidae, Underground bioacustics, Fossorial rodents, Vocal development

Funding: The authors received no funding for this work.

==============================
Tuco-tucos (Ctenomys spp.) are subterranean rodents that produce territorial, high intensity long-range vocalizations (LRVs) of broadband and low frequency that are essential for long-distance communication between individuals in different tunnel systems. Despite their importance, the development of LRVs remains poorly understood. In adult Anillaco Tuco-Tucos (Ctenomys sp.) the LRV is composed by two types of syllables (series and individual notes) that are repeated a variable number of times. We studied the development of the LRVs in eight juveniles of the Anillaco Tuco-Tuco ranging from 14–28 to 104–118 days after birth. We (1) tested whether the syllables followed any of three alternative developmental modes (retention of juvenile vocalizations, modification of juvenile precursors or de novo appearance in adults), (2) evaluated the development of structural and acoustic features of syllables, and (3) tested the prediction that juveniles should produce a greater proportion of atypical series in precursors of the LRV than adults, due to lack maturation and/or precise coupling of neuromuscular and anatomical structures. The LRV of the Anillaco Tuco-Tuco exhibited a mixed developmental mode: while series developed from juvenile precursors whose acoustic features gradually approached those of adults, individual notes appeared later in the ontogeny and de novo with acoustic features indistinguishable from those of adults. The number of series per vocalization increased through development and varied from one to 25 in juvenile males and from one to six in juvenile females. The structure of the most common series type (triad) did not exhibit ontogenetic changes and was present as such at the onset of the emission of vocalizations. On the contrary, acoustic features of juvenile triad notes changed with age in both sexes (duration 90% increased through development, while bandwidth 90% and peak frequency decreased). Furthermore, juveniles emitted a higher proportion of atypical series than adults (7.4% vs. 0.3%), as expected in the development of any complex behavior that requires practice to be mastered. The maturation of the LRV occurred well before the sexual maturation, presumably due to the protracted time needed to acquire or build a burrow system long before mating is possible. We propose that protracted vocal development is another component in the slow developmental strategy of Ctenomys and subterranean rodents in general.

Introduction

Like many other aspects of biological systems, vocalizations follow a developmental trajectory during ontogeny. In principle, adult vocalizations could develop following three alternative modes. Adults may just retain the vocalizations of juveniles (Winter et al., 1973; Campbell et al., 2014), vocalizations may develop through the modification of juvenile precursors that eventually reach adult features (Clemmons & Howitz, 1990; Grimsley, Monaghan & Wenstrup, 2011), or they may appear de novo in adults without the need of homologous juvenile precursors (Campbell et al., 2014). Vocalizations that develop through precursors may follow two non-mutually exclusive developmental pathways: vocal learning, that involves the modification of vocalizations based on the copy of sounds from an external auditory input (Tyack, 2016); and vocal tract maturation, that involves the capacity of juveniles to generate adult-like vocalizations by the tuning of an innate motor program (Winter et al., 1973; Tyack, 2016).

The acquisition of adult vocal patterns requires practice, since it is necessary to synchronize the neuromuscular system and anatomical structures, either to match the vocal output to those signals acquired through external input or to achieve the successful deployment of instructions contained in the internal innate motor program (Boughman & Moss, 2003). The lack of maturation and/or precise coupling between the morphology and the neuromuscular system of the vocal apparatus leads to the production of atypical vocalizations. Consequently, regardless of the developmental pathway followed, juveniles are expected to produce a greater proportion of atypical vocalizations during their practice period than adults during the emission of their mature vocalization (Konishi, 1965).

Tuco-tucos (Ctenomys spp.) are subterranean rodents that produce long-range vocalizations (LRVs) of high intensity, broadband and low frequency. These vocalizations are essential for long-distance communication between individuals in different tunnel systems (Schleich & Busch, 2002; Amaya et al., 2016). Multiple lines of evidence suggest that LRVs are territorial signals that facilitate the maintenance of individual territories and minimize aggressive encounters, especially between males (Francescoli, 1999; Schleich & Busch, 2002; Amaya et al., 2016). Despite their behavioral importance, the LRVs of adults of only three of the more than 60 recognized species of Ctenomys (Bidau, 2015) have been characterized in some detail (Francescoli, 1999; Schleich & Busch, 2002; Amaya et al., 2016). Juvenile LRVs were mentioned in C. talarum (Zenuto, Antinuchi & Busch, 2002) and C. mendocinus (Camín, 2010), but there is no detailed information on the developmental modes, pathways or developmental times of LRVs. Learning how and when the LRVs reach maturity can improve our understanding of how life-history traits interact and influence each other in fossorial rodents. For example, since burrows are energetically expensive resources (Vleck, 1979; Vleck, 1981) it is conceivable that solitary species of Ctenomys secure or build their own burrow system before mating. This would imply the need of having mature LRVs to help defend the burrows during the process of acquiring a mate, even before reaching sexual maturity.

The LRV of adult Anillaco Tuco-Tucos (Ctenomys sp.) is composed of two types of syllables: series and individual notes, that are repeated a variable number of times in each vocalization (Fig. 1). All LRVs contain series, but not always include individual notes (Amaya et al., 2016). Series are composed by two sound types: notes, loud sounds that initiate and finish the series; and soft-notes, softer sounds that are sandwiched by notes. Depending on the number of notes and soft notes, series can belong to any of three series-patterns: dyads (two notes, one soft note), triads (three notes, two soft notes), or tetrads (four notes, three soft notes). Triads constitute the most common series pattern. However, adults occasionally produce atypical series by adding a soft-note before the initial note. The individual notes are single notes emitted at a faster rate in succession and are always given after the series (Fig. 1) (Amaya et al., 2016).

Figure 1 General structure of the long-range vocalization (LRV).

(A) General structure of the long-range vocalization (LRV) of the Anillaco Tuco-Tuco (Ctenomys sp.). (B) Syllable types (series and individual notes). Triad series are composed by three notes (N1, N2 and N3) and two soft-notes (Sa and Sb), and individual notes are simple sounds. Waveforms are shown above and corresponding spectrogram below in both figures.

Structural and quantitative acoustic traits of vocalizations may experience different developmental trajectories (Elowson, Snowdon & Sweet, 1992; Brittan-Powell, Dooling & Farabaugh, 1997; Hammerschmidt et al., 2000; Hammerschmidt, Jürgens & Freudenstein, 2001; Grimsley, Monaghan & Wenstrup, 2011; Campbell et al., 2014). Structurally, triads are the most common series pattern in adults of the Anillaco Tuco-Tuco. These triads could conceivably develop through the initial emission of separate notes and soft-notes, passing to their joint emission in the shape of monads, subsequently leading through the repetition of this basic structure to the creation of the other types of series composed by more notes and soft-notes. Alternatively, the structure of series could be a hard-wired feature that would experience no ontogenetic changes, with triads and other series types existing since early ontogenetic stages. Likewise, quantitative acoustic features of LRVs such as frequency and duration could develop gradually or may simply appear with adult parameters. Since body size in most mammals is inversely correlated with peak frequency and juveniles tend to produce less precise vocalizations (Morton, 1977; Owings & Morton, 1998), we expect peak frequency and bandwidth to decrease and length of syllable types to approach adult values.

This paper aims to describe the developmental trajectory of the LRVs in the Anillaco Tuco-Tuco based on recordings of captive-reared pups, in order to (1) discuss which developmental mode (juvenile retention, juvenile precursor, or de novo appearance) explain the ontogeny of the two types of syllables of the LRV; (2) evaluate the development of structural and quantitative acoustic features of syllables in LRVs and compare them with those in adults; (3) test the prediction that juveniles should produce a greater proportion of atypical series in precursors of the LRV than adults; and (4) discuss the relationship between the development of LRVs and other life-history traits.

Materials and Methods

Experimental animals

We studied the development of the LRV in eight juveniles (five males and three females) of the Anillaco Tuco-Tuco. Taxonomy of this species is still cursory and multiple lines of evidence indicate that our study area is occupied by a single and yet unidentified Ctenomys species that is closely allied to species described from similar dry habitats in neighboring provinces (Amaya et al., 2016).

Juveniles of the Anillaco Tuco-Tuco were trapped at Anillaco, La Rioja, Argentina (28°48′50″S, 66°55′54″O; 1,365 m asl) in November and December yearly between 2013 and 2016. Immediately after trapping, juveniles were carried to the Chronology and Ethology lab in the CRILAR to a room (2.5 m × 4 m) with a fluorescent light source (36 W) controlled by a mechanical timer (TBCin-China) providing a light-dark cycle of 12:12. Each juvenile was housed in a transparent glass enclosure (90 × 40 × 40 cm, see Amaya et al., 2016) with abundant wood chips on the floor and PVC tubes used as burrows. Food was offered ad lib and consisted of sweet potatoes, carrots, sunflower seeds and natural food items collected weekly from the field (Larrea sp. and Opuntia sp.).

Juveniles were acoustically isolated from adults during their development in captivity. Male 1 and female 1 were kept alone in 2013 and 2014 respectively; while males 2 and 3, and female 2 were kept together in 2015; and males 4 and 5, and female 3 were kept together in 2016. Juveniles were weighed four times during our study (when trapped and three times at intervals of 30–35 days in captivity; Table 1) and sexed a posteriori because the genitalia were not mature at the time of capture.

Table 1 Long-range vocalizations emitted by juveniles during vocal development.

Structural features of the long-range vocalizations (LRVs) emitted by eight juveniles (five males and three females) of the Anillaco Tuco-Tuco (Ctenomys sp.) during four sampling weeks over 12 study weeks. For each juvenile individual we indicate weight (W) (g), estimated age in days (D), and recording time in hours (R). For each vocalization recorded (Roman numerals) we indicate the number of each normal series-pattern (triads in bold font; dyads in normal font; and tetrads with underline), and the number of individual notes per vocalization (underlined). Deceased (†), no vocalization (−).

Juveniles individual	Week 1	Week 4	Week 8	Week 12	
	W	D	R	Vocalizations	W	D	R	Vocalizations	W	D	R	Vocalizations	W	D	R	Vocalizations	
				I	II	III	IV				I	II	III	IV				I	II	III	IV				I	II	III	IV	
Male 1	73	28	22	1	1	1	–	101	58	24	1	1	1
1	–	125	88	18	2
3	2
3	1
6	–	162	118	24	2
4	10	2
18
26	–	
Male 2	47	18	22	1	1	–	–	85	48	20	1	1	1
2	–	119	78	24	1
2	2
3	6
4		156	108	24	2
3	2
16	8
12
31	–	
Male 3	45	17	20	1	1	–	–	80	47	22	1	2	–	–	117	77	20	4	7	–	–	†	†	†	†	†	†	†	
Male 4	71	27	24	1	1	3	–	97	57	24	1
1	3	5	–	123	87	28	2
5	13	2
9	–	158	117	28	13	4
21
22	–	–	
Male 5	49	19	24	1	1	–	–	81	49	24	2	4	–	–	111	79	28	1
1	2
5	4
3	–	150	109	28	5	1
6	15	–	
Female 1	39	14	22	1	1	1	–	68	44	24	1	1	1
1	–	95	74	22	1	2	3	–	125	104	20	5	6	–	–	
Female 2	59	23	24	1	2	1	–	85	53	24	1	1
1	–	–	110	83	28	1	3	–	–	142	113	28	2	3	6	–	
Female 3	55	21	20	–	–	–	–	70	51	18	–	–	–	–	92	81	22	–	–	–	–	115	101	24	–	–	–	–	

Since we captured juveniles of unknown biological age, we estimated their absolute ages at trapping by comparing their weights to those of juvenile C. mendocinus of known age (Camín, 2010). Adults of the Anillaco Tuco-Tuco are 10% heavier than adults of C. mendocinus (Camín, 2010; Amaya et al., 2016). Thus, we assumed a linear age-weight relationship such that 10% heavier juveniles of the Anillaco Tuco-Tuco where aged as equivalent to 10% lighter juveniles of C. mendocinus (e.g., at day 14 juveniles of C. mendocinus weigh on average 33 g, while at this age the weight of juvenile Anillaco Tuco-Tucos was estimated to be around 36.3 g). The minimum estimated age at trapping was 14 days (female 1; Table 1) and the maximum 28 days (male 1; Table 1). To conduce statistical comparisons we merged data of juvenile males and females in four age classes: 14–28, 44–58, 74–88, and 104–118 days after birth.

All procedures followed the guidelines of the American Society of Mammologists for the use of wild mammals in research (Sikes & Gannon, 2011). All experiments were performed at the CRILAR and were authorized by the Environmental Department of La Rioja (exp. P40015215). All individuals were returned to their natural environment after the study was complete.

Recording of vocalizations

We studied the development of LRVs of eight juveniles of Anillaco Tuco-Tuco. Each juvenile was recorded during four 6–8 hr sessions in alternate days within a week every four weeks (i.e., a week of recording with four sessions was separated by three weeks without recording). Sessions began during the first week that individuals were trapped and finished after twelve weeks. Thus, each individual was recorded in 16 sessions spread along 12 weeks, except for male 3 which died in the ninth week and was recorded in only 12 sessions.

Recordings were made from outside the transparent glass enclosure using a Zoom H4n digital hand recorder system with built-in microphones (sample rate of 44.1 kHz and 24 bit depth) mounted on a tripod with the microphones facing the enclosure. The gain setting of the recorder was the same for all recordings. Our recording protocol allowed us to detect and record the emission of the LRVs, which occur at a low rate and unpredictably. All the recording sessions were made during the night (between 22:00 and 6:00 hs Argentina time, GMT -5), due the low level of ambient noise and to the high nocturnal activity of the species in laboratory conditions (Tachinardi et al., 2014). In total, we recorded five males for 448 h in 76 sessions and three females for 276 h in 48 sessions.

We searched for LRVs in recordings by manually examining on-screen spectrograms built with Raven pro 1.4 (http://www.birds.cornell.edu/Raven). We used the following spectrogram parameters to prioritize resolution in time to be able to easily detect vocalizations: Window: Hann; size: 256 samples (= 5,8 ms); 3 dB bandwith-filter: 248 Hz; Time grid-overlap: 50%; hop size: 128 samples (= 2,9 ms); Frequency grid–DFT size: 256 samples; grid spacing: 172 Hz.

Acoustic characterization of the long-range vocalization

We analyzed all juvenile vocalizations to quantify the number of series and individual notes emitted per vocalization and selected a subset to acoustically characterize them. Of all the types of series in juveniles, we measured only the triads because they are the most common and characteristic series type of the LRV in adults of these species (Amaya et al., 2016). We characterized one triad and three individual notes per vocalization. In total, we analyzed 47 triads in males and 19 in females; and nine individual notes in males (see Table S1 for sampling in each age class). To compare triads and individual notes of juveniles with those of adults, we characterized them in five adult males and three adult females (one triad and three individual notes per adult individual), using adult recordings from Amaya et al. (2016). We measured duration 90% (ms), bandwidth 90% (Hz) and peak frequency (Hz) of each triad note and individual notes in juveniles and adults. We calculated single duration 90%, bandwidth 90% and peak frequency values for each individual in each vocalization by averaging the values of the three triad notes and the three individual notes. These acoustic parameters were defined as follows: 1-duration 90%, the difference between time 5% (time that divides the selection border in two time intervals containing 5% and 95% of the energy) and time 95% (time that divides the selection border into two time intervals containing 95% and 5% of the energy); 2-bandwidth 90%, the difference between frequency 5% (frequency that divides the selection into two frequencies intervals containing 5% and 95% of the energy) and frequency 95% (frequency that divides the selection border into two frequencies intervals containing 95% and 5% of the energy); and 3-peak frequency, the frequency at which the peak energy is produced within the selection border.

We delimited the triad notes and individual notes for measurements of duration 90% (ms), bandwidth 90% (Hz) and peak frequency (Hz) as follows. For triads, we delimited an initial selection border A (400 ms and 0–22.05  kHz) (Fig. 2A) that was subdivided in 20 equal selection borders B (20 ms and 0–22.05 kHz) (Fig. 2B), and we defined a selection border C (40 ms and 0–22.05 kHz) delimiting each triad note by choosing the two adjacent selection borders B with the highest energy values (Fig. 2B). We performed the same procedure for individual notes, where an initial selection border D (80 ms and 0–22.05 kHz) (Fig. 2C) was subdivided into 4 equal selection borders E (20 ms and 0–22.05 kHz) (Fig. 2C), and we defined a selection border F (40 ms and 0–22.05 kHz) delimiting each individual note by choosing the two adjacent selection borders E with the highest energy values (Fig. 2D).

Figure 2 Acoustic characterization of the long-range vocalization (LRV).

Delimitation of selection borders used for the quantitative acoustic characterization of triads (A and B) and individual notes (C and D) of the long-range vocalization (LRV) during the vocal development of juveniles and in adults of the Anillaco Tuco-Tuco (Ctenomys sp.). Waveforms are shown above and corresponding spectrogram below in all figures. (A) Selection border A delimiting a triad. (B) Selection borders B and selection border C delimited by the two adjacent selection borders B with the highest energy values. (C) Selection border D delimiting an individual note. (D) Selection borders E and selection border F delimited by the two adjacent selection borders E with the highest energy values. We measured duration 90%, bandwidth 90% and peak frequency in selection borders C (in triads) and F (in individual notes).

All acoustic measurements were made with Raven Pro 1.4. We used different spectrograms parameters that allowed us to obtain accurate time and frequency domain measurements. For duration 90% we used the spectrogram parameters used to search for LRVs in recordings (see above). For bandwidth 90% and peak frequency the spectrogram parameters were: Window: Hann, size: 1,024 samples (=52.4 ms), 3 dB bandwith-filter: 27.4 Hz; Time grid–overlap: 50%, hop size: 512 samples (=26.2 ms); Frequency grid–DFT size: 4,096 samples, grid spacing: 4.7 Hz. All recordings were band-pass filtered between 80–5,000 Hz in Raven Pro 1.4 to eliminate sources of disturbance and distortion in acoustic measurements as in Amaya et al. (2016).

Series-patterns and atypical series

We analyzed all juvenile vocalizations to quantify the occurrence of each of the three normal series-patterns (dyads, triads and tetrads). We first identified the series-patterns in the corresponding waveform and spectrogram by counting the number of notes and soft-notes per series, and then counted the number of series-patterns per vocalization. Following the same procedure, we quantified the occurrence of atypical series in juvenile vocalizations. We considered a series to be atypical if it had a soft-note before the note 1 in dyads, triads and tetrads or if it was composed by a single note followed by a soft-note (i.e., it was a monad) (Fig. 3).

Figure 3 Abnormal series.

Abnormal series produced by juveniles of the Anillaco Tuco-Tuco (Ctenomys sp.) during the development of the long-range vocalization (LRV). Waveforms are shown above and corresponding spectrogram below in all figures. (A) Abnormal triads. (B) Abnormal dyads. (C) Abnormal tetrad. (D) Monad. Previous soft-note (P S); note (N); soft-note (S). Numbers indicate the corresponding note or soft-note in the series.

In order to test the prediction that juveniles produce proportionally more atypical series than adults, we compared the proportion of atypical series in adults using data from Amaya et al. (2016) to the proportion of atypical series in juveniles. To calculate these proportions (expresed as %) we counted the number of atypical series and divided these counts by the total number of series in juveniles and in adults. In total, 325 juvenile series were recorded for the present study and compared against 703 adult series from Amaya et al. (2016).

Statistical analyses

To evaluate if the acoustic features of juvenile male triad notes and if the number of series per vocalization changed during their development, we performed a non-parametric repeated measures analysis of variance (Friedman test) of duration 90%, bandwidth 90%, peak frequency and number of series per vocalization across four age classes (14–28, 44–58, 74–88 and 104–118 days after birth). Post-hoc analyses were conducted using Bonferroni multiple comparison tests. To compare at which point acoustic features of juvenile male triad notes reached adult parameters during their vocal development, we performed a Wilcoxon Signed-Rank test comparing duration 90%, bandwidth 90%, and peak frequency of triad notes of each juvenile age class against values in adult males. Few vocalizations of juvenile males contained individual notes and all appeared at the 104–118 dab age class. Thus, we visually compared values of acoustic parameters of juvenile individual notes against those in adults.

Due to the low number of juvenile females, comparisons among age classes and between age classes and adults were restricted to visual comparisons of acoustic parameters along their ontogeny.

For each variable analyzed, we used the average value of measurements taken from each individual within each age class. All analyses were carried out in R (R Core Team, 2017) with alpha set at 0.05.

Results

Seven juveniles of the Anillaco Tuco-Tuco (five males and two females) produced 71 LRVs in the 12 study weeks; while an eighth juvenile (female 3) did not produce any vocalization (Table 1). The two syllable types of LRVs, series and individual notes, appeared sequentially in the vocal development (Fig. 4). Series appeared first and were produced by juveniles as young as 14 days old (39 g). Individual notes were produced considerably later in the development, when juveniles were as young as 108 days old (156 g) (Fig. 4, Table 1). Accordingly, most vocalizations were composed only by series (95.77%, N = 68), and fewer vocalizations were composed by series and individual notes (4.22%, N = 3) (Table 1).

Figure 4 Development of the long-range vocalization (LRV).

Example of the development of the long-range vocalization (LRV) in a male of the Anillaco Tuco-Tuco (Ctenomys sp.) during four sampling weeks over 12 study weeks. During the sampling period this individual (male 2) gave 11 vocalizations (numbered boxes). Note the increase in the number of series per vocalization and the late appearance of individual notes. These two features are representative of the development of the LRV in this species. For each sampling week we indicate weight (g) and estimated age (days after birth).

Development of series

The seven juveniles that vocalized produced series (Table 1). The number of series per vocalization increased through development and varied from one to 25 in juvenile males (Table 1) and from one to six in juvenile females (Table 1). The first vocalizations of all juveniles contained a single series, and the highest numbers of series were recorded in juvenile males between 108 to 118 days old, and in juvenile females between 104 to 113 days old (Table 1).

Figure 5 Developmental changes in acoustic features in the long-range vocalization.

Developmental changes in acoustic features of triad notes and individual notes in the long range vocalization (LRV) of the Anillaco Tuco-Tucos (Ctenomys sp.) and comparison to adult values. All data from juveniles and adults in Table S1. (A) Duration 90%. (B) Bandwidth 90%. (C) Peak frequency.

Juveniles produced the three series-patterns present in the adults. Triads were by far the most common series-pattern (82.76%, N = 269), followed by dyads (16.61%, N = 54), and tetrads (0.61%, N = 2) (Table 1). Triads were produced by all vocalizing juveniles, dyads were produced by all juveniles (except male 3 which died prematurely), and tetrads were produced by only two juveniles (female 1 and female 2) (Table 1).

All three acoustic features of triad notes and the number of series per vocalization changed along the ontogenetic trajectories of juvenile males and females (Figs. 4 and 5, Table 2 and Table S1). In juvenile males, duration 90% of triad notes exhibited a significant directional increase through development (Friedman test; n = 5, p < 0.04) (Fig. 5, Table 2 and Table S1); bandwidth 90% and peak frequency of triad notes exhibited a significant directional decrease during the development (Friedman tests; bandwidth 90%: n = 5, p < 0.01; and peak frequency: n = 5, p < 0.01) (Fig. 5, Table 2 and Table S1). Finally, the number of series per vocalization increased through development (Friedman test; n = 5, p < 0.007) (Fig. 5, Table 2 and Table S1). Although we could not perform statistical comparisons in females, our data show that through development duration 90% of triad notes increased (Fig. 5, Table 2 and Table S1), bandwidth 90% and peak frequency of triad notes decreased (Fig. 5, Table 2 and Table S1), and the number of series per vocalization increased (Table 1). Thus, juvenile males and females showed similar ontogenetic changes in their series.

Table 2 Statistical analyses.

Quantitative comparisons of acoustic parameters (duration 90%, bandwidth 90% and peak frequency) of triad notes and number of series in the long-range vocalizations (LRVs) between four age classes of juvenile males of the Anillaco Tuco-Tuco (Ctenomys sp.) and acoustic parameters of triad notes of each juvenile age class against adults of the corresponding sex. Friedman test and Bonferroni post hoc-test results comparing acoustic parameters of triad notes and number of notes per vocalization between different juvenile males age classes; shared letters indicate no statistical differences and different letters indicate significant statistical differences between age classes. Wilcoxon Signed-Rank test results compare acoustic parameters of all triad notes of different age classes of juvenile males against adults; P-values in bold indicate statistically significant differences between an age class in comparison to adult values. Data presented is mean ± sd (range). See Table S1 for each juvenile and adult values.

	Statistical test	Series notes parameters	Juvenile age classes (days after birth)	Adults	
			14–28	44–58	74–88	104–118		
Males	Friedman test (Bonferroni post-hoc test)	Duration 90% (ms)	A	B	B	B	–	
Bandwidth 90% (Hz)	A	AB	B	B	–	
Peak frequency (Hz)	A	AB	B	B	–	
Series per vocalization	A	AB	B	C	–	
N	5	5	5	4	–	
Wilcoxon Signed-Rank test	Duration 90% (ms)	p < 0.001
28.2 ± 2.1
[25.1–31.9]	p < 0.3
30.5 ± 2.2
[27.9–33.8]	p < 0.6
31.9 ± 1.5
[29.9–34.8]	p < 0.8
30.8 ± 2.0
[27.0–32.9]	–
31.4 ± 1.7
[29.3–34.6]	
Bandwidth 90% (Hz)	p < 0.001
832.6 ± 771.5
[234.4–2637.8]	p < 0.01
479.1 ± 186.8
[201.0–778.8]	p < 0.06
268.8 ± 55.5
[186.6–351.7]	p < 1
207.6 ± 82.5
[96.9–305.1]	–
192.7 ± 67.8
[105.5–257.8]	
Peak frequency (Hz)	p < 0.001
376.1 ± 193.4
[222.5–832.6]	p < 0.001
254.4 ± 47.1
[208.2–367.2]	p < 0.03
217.4 ± 36.3
[157.9–269.1]	p < 0.1
207.6 ± 82.5
[96.9–305.1]	–
187.5 ± 15.6
[164.1–210.9]	
N	5	5	5	4	5	
Females	–	Duration 90% (ms)	26.1 ± 1.3
[25.2–28.1]	29.7 ± 1.9
[27.1–31.9]	31.9 ± 0.7
[31.0–32.9]	31.7 ± 0.1
[30.0–32.9]	31.3 ± 0.1
[29.3–32.0]	
Bandwidth 90% (Hz)	1545.9 ± 769.8
[452.2–2128.2]	497.7 ± 63.3
[426.1–556.3]	337.9 ± 124.5
[215.3–463.0]	260.2 ± 65.5
[168.7–315.8]	293.0 ± 57.4
[246.1–363.3]	
Peak frequency (Hz)	508.7 ± 139.4
[337.4–674.7]	322.2 ± 35.6
[269.5–348.1]	315.8 ± 99.6
[244.1–463.0]	274.7 ± 59.3
[210.9–333.8]	249.0 ± 17.5
[234.4–269.5]	
N	2	2	2	2	3	

Acoustic features of triad notes of juvenile males progressively approached features of adult males and achieved mature features at different developmental stages. Pairwise comparisons (Wilcoxon Signed-Rank test) of duration 90%, bandwidth 90% and peak frequency of juvenile and adult males showed significant differences at different ages (Fig. 5, Table 2 and Table S1). Thereby, acoustic features of triad notes passed through a juvenile developmental stage in which they were different from adults before reaching adult values (Fig. 5, Table 2 and Table S1). We were unable to compare the acoustic features of triad notes between juvenile and adult females statistically. However, visual inspection of average acoustic values show that triad notes of juvenile females had shorter duration 90% and higher bandwidth 90% and peak frequency than those in adult females. Thus, just like juvenile males, juvenile females seem to gradually acquire acoustic features of triad notes of adult females during their vocal development (Fig. 5, Table 2 and Table S1).

Atypical series in juveniles and adults

Juveniles emitted two classes of atypical series during the development of the long-range vocalization. In the first class of atypical series a soft note preceded the first series note, and included atypical triads (Fig. 3A), dyads (Fig. 3B) and tetrads (Fig. 3C). In the second class, the series was conformed by a single monad that consists of a single note followed by a soft-note (Fig. 3D). Monads are unknown in adults (see Amaya et al., 2016).

The percentage of atypical to normal series was more than an order of magnitude larger in juveniles than in adults. Juveniles produced 7.38% of atypical series (eight atypical triads, 11 atypical dyads, three atypical tetrads, and four monads vs. 299 normal series) while in adults only 0.28% of the series were atypical (one atypical triad and one atypical dyad vs. 701 normal series).

Development of individual notes

Only three of the five males and none of the two females that vocalized gave individual notes. These notes were given by males 1, 2 and 4 when they had between 156–162 g and were 108–118 days old (Table 1); while male 3 died at an early age, male 5 approached this weight and was within this age range but never gave individual notes (Table 1). Individual notes were recorded in only a single vocalization in each male, and ranged in number from 22–31 notes always emitted after a sequence of series and never in isolation (Table 1). Visual comparisons of average values of duration 90%, bandwidth 90% and peak frequency of individual notes (Fig. 5, Table S1) show that acoustic parameters of individual notes of juveniles were indistinguishable from those of adults (Fig. 5 and Table S1).

Discussion

In this paper we have shown that the long-range vocalization (LRV) of the Anillaco Tuco-Tuco (Ctenomys sp.) exhibits a mixed developmental mode in which the two syllables types of the vocalization develop differently: while series developed from juvenile precursors whose acoustic features gradually approached those of adults, individual notes appeared later in the ontogeny and de novo, with acoustic features indistinguishable from those of adults. The most common series type did not exhibit ontogenetic changes in its structure: triads appeared structurally as such at the onset of the emission of vocalizations at early developmental stages. Finally, we found that juveniles emit a higher proportion of atypical series than adults, as expected in the development of any complex behavior that requires practice to be mastered.

Development of the long-range vocalization

The LRV in the Anillaco Tuco-Tuco exhibits a mixed developmental mode, since it includes a syllable type (series) that develops gradually from structurally conserved ontogenetic precursors in the juveniles and another syllable type (individual notes) that appears de novo with adult features. This sort of mixed development is novel and has not been reported in the vocalizations of other rodents. For example, while the vocalization of Mus musculus develops from precursors of the syllables that increase in duration and decrease in fundamental frequency (Grimsley, Monaghan & Wenstrup, 2011), the acoustic characteristics of the notes in Scotinomys spp. do not experience ontogenetic changes (Campbell et al., 2014) and are therefore the product of retention of juvenile vocalizations.

Early in the ontogeny, juveniles of the Anillaco Tuco-Tuco produce the three series-patterns present in adults (dyads, triads and tetrads). The subsequent lack of structural changes in the structure of series is interesting since a priori, other developmental mechanisms could have occurred. For example, series could have developed by the sequential addition of simple notes and soft-notes, subsequently coupled to create the series-patterns. Triads were the most common series-pattern in adults (Amaya et al., 2016) and in juveniles (this work). We interpret the early appearance of triads and their ubiquitous presence as evidence of a substantial genetic makeup in the structural composition of series in Ctenomys. In fact, other series-patterns predominate in other species of tuco-tucos. For example, monads are characteristic of Ctenomys talarum (Schleich & Busch, 2002; J Amaya & J Areta, 2018, unpublished data) and dyads are characteristic of Ctenomys mendocinus (Amaya et al., 2016; J Amaya & J Areta, 2018, unpublished data). Further studies in these species could help clarify the genetic contribution to the determination of the structure of series-patterns in Ctenomys.

Although triads appear early in the development of juveniles, their acoustic features differ from those of adults. Juvenile precursor triad notes increase duration; and decrease bandwidth and peak frequency gradually until reaching the values of adult triads. Thereby, juvenile triad precursors develop into adult triads by gradual changes in their acoustic parameters but without changing their basic structure.

The lower degree of morphological and neuromuscular maturation and tuning in the vocal apparatus of juvenile in comparison to adult tuco-tucos may account for the higher proportion of atypical series in juvenile individuals. This explanation may be more general and could also apply to juveniles of two species of singing mice (Scotinomys teguina and S. xerampelinus). Five types of notes were described in juvenile Scotinomys spp. (Campbell et al., 2014); one very frequent note (note A) exists at all ages and is almost the only note sang by adults, while four less frequent notes are more frequent in the first 12 days of life and become almost non-existent after 30 days (Campbell et al., 2014). We speculate that these four types of less frequent notes are not different notes per se, but that they could simply represent atypical renditions of the note A. Hence, their decrease in frequency could be explained by the improved performance of the vocal tract and associated neuromuscular apparatus through development.

Individual notes of LRVs appear late, de novo and with acoustic parameters indistinguishable from those of adults, showing a developmental mode that is strikingly different to that of series. The first individual notes are given by juveniles when the series have already acquired adult acoustic characteristics through maturation from precursors. This suggests that the appearance of individual notes with their definitive acoustic characteristics without the existence of ontogenetic precursors may be possible because the neuromuscular system and the vocal apparatus have acquired the necessary “fine tuning” through practice of the series during their development. The passive maturation of the individual notes might be, at least in part, a co-product of the active maturation of the series.

The LRVs in the Anillaco Tuco-Tuco exhibit a large variety of syntactic patterns (Amaya et al., 2016). These syntactic patterns undergo some general and predictable ontogenetic changes that include, first an increase in both the number of series and series-patterns per vocalization, and then an increase in syllable types given by the late appearance of individual notes. The increment in vocalization length and the late appearance of individual notes might be widespread in Ctenomys. These patterns appear to occur also in the vocal development of C. mendocinus, where ”short calls” presumably conformed only by series occur at 33 days and ”long calls” presumably conformed by series and individual notes occur at 70 days (Camín, 2010). In contrast, the highly stereotyped pattern of Scotinomys adult song appears de novo in the first vocalization of juveniles after a period of vocal inactivity (Campbell et al., 2014).

Vocalizations of rodents appear to develop by maturation of the vocal tract and not by vocal learning since adult vocalizations develop normally without the need of an external auditory model from which to copy (Scherrer & Wilkinson, 1993; Grimsley, Monaghan & Wenstrup, 2011, but see Arriaga, Zhou & Jarvis, 2012). In this paper we demonstrated that juveniles of the Anillaco Tuco-Tuco develop normal LRVs without an adult external auditory input between 14 and 108 days. This would indicate that the development of this vocalization does not depend on vocal learning processes (Efremova et al., 2011). However, juveniles were exposed to adult LRVs from birth until they were captured, and Ctenomys are born with an open auditory meatus (Zenuto, Antinuchi & Busch, 2002; Camín, 2010). Thus, it remains possible that adult vocalizations heard at an early age have served as an auditory model for the subsequent vocal development. More thorough future tests on the importance of vocal learning in the development of LRVs in Ctenomys should focus on evaluating if deafened juveniles develop normal LRVs when devoid from acoustic self-stimulation (Kroodsma & Konishi, 1991; Campbell et al., 2014).

Life-history traits and territorial vocalizations in subterranean rodents

Subterranean rodents exhibit slow life-histories, which are characterized by slow speed of tissue growth due to a low basal metabolic rate selected for by the high energetic demands of burrowing (Henneman, 1983; Henneman, 1984; Hofman, 1983; Glazier, 1985) leading to delayed sexual maturity (Busch et al., 2000). In Ctenomys talarum and C. mendocinus gestation lasts around 95 days (Zenuto, Antinuchi & Busch, 2002; Camín, 2010); sexual maturity takes 180–240 days in C. talarum (Busch et al., 2000) and presumptively adult vocalizations have been reported long before sexual maturity at 65–70 days (Zenuto, Antinuchi & Busch, 2002) and the same pattern is evident in C. mendocinus which reaches sexual maturity around 180–270 days (Camín, 2010) and supposed vocal maturity at 70 days (Camín, 2010). We lack precise information on the age of sexual maturity in the Anillaco Tuco-Tuco, but the patterns described for C. talarum and C. mendocinus suggest that vocalizations mature well in advance of sexual maturity in Ctenomys. The solitary bathyergids, Bathyergus janetta, B. suillus and Georychus capensis exhibit a moderate gestation period of around 50 days and presumably delayed sexual maturation (Jarvis, 1969; Bennett & Jarvis, 1988; Bennett et al., 1991; Bennett, Faulkes & Molteno, 2000). In B. janetta and B. suillus, juveniles begin to produce seismic signals (foot-drumming) at 80 days, after dispersion at 60–65 days from the maternal burrow system; while in Georychus capensis juveniles produce foot-drumming at 50 days of age before dispersion at 55–60 days of age (Bennett & Jarvis, 1988). Dispersion in these bathyergids, occurs when they have the capacity to dig and protect their own burrow systems (Bennett & Jarvis, 1988). Thus, territorial acoustic signals of solitary ctenomyids and bathyergids seem to mature approximately at the time of dispersion but long before sexual maturity.

The protracted development of the LRV of the Anillaco Tuco-Tuco (ca. 110–120 days) herein reported contrasts markedly with the short developmental times of vocalizations of rodents that live in the surface such as Scotinomys (35 days; Campbell et al., 2014) and Mus musculus (at 13 days individuals had some adult vocal characteristics, Grimsley, Monaghan & Wenstrup, 2011). The altricial Ctenomys contrast with the precocial Scotinomys in which gestation takes 30 days (Hooper & Carleton, 1976) and which reach sexual maturity after 30–40 days concomitantly with the maturation of adult vocalizations (Hooper & Carleton, 1976; Campbell et al., 2014).

We propose that the protracted vocal development is another component in the slow developmental strategy of Ctenomys and subterranean rodents in general. The underground burrow systems have a central role in the ecology of subterranean rodents (Busch et al., 2000) and their defense is of key importance. Burrows are costly resources that serve for breeding and that are defended by the use of long-range vocalizations in Ctenomys (Amaya et al., 2016). The maturation of vocalizations in advance of sexual maturation might be explained by the protracted time needed to acquire or build a burrow system long before mating is possible.

Supplemental Information

Table S1 Acoustic parameters of triad notes and individual notes

Acoustic parameters (duration 90%, bandwidth 90%, and peak frequency) of triad notes and individual notes of four different age classes (days after birth) of juvenile and adult males (n = 5) and females (n = 3) of the Anillaco Tuco-Tuco (Ctenomys sp.). Data presented is mean ± sd (range) (sample size).

Click here for additional data file.

Data S1 Raw data

Click here for additional data file.

We are grateful to Pablo Lopez, Tatiana Sanchez, Johana Barros and Veronica Valentinuzzi for help with care and feeding of the tuco-tucos in the lab. Hynek Burda, Elke Zimmermann and two anonymous reviewers provided critical reviews and suggested numerous venues for improvement.

Additional Information and Declarations

Competing Interests

Author Contributions

Animal Ethics

Data Availability

The authors declare there are no competing interests.

Juan Pablo Amaya conceived and designed the experiments, performed the experiments, analyzed the data, contributed reagents/materials/analysis tools, wrote the paper, prepared figures and/or tables.

Juan Ignacio Areta conceived and designed the experiments, analyzed the data, contributed reagents/materials/analysis tools, wrote the paper, prepared figures and/or tables, reviewed drafts of the paper.

The following information was supplied relating to ethical approvals (i.e., approving body and any reference numbers):

All experiments were performed at the CRILAR in Anillaco and were authorized by the Environmental Department of La Rioja (File number. P40015215).

The following information was supplied regarding data availability:

The raw data has been provided as a Supplemental File.

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
