# Peer review of "Ontogeny of long-range vocalizations in a Neotropical fossorial rodent: the Anillaco Tuco-Tuco (Ctenomys sp.)"

_PeerJ, doi:10.7717/peerj.4334_

## Round 0.1 · original submission · Major Revisions

All three referees and myself found that the manuscript may have the potential for publication in PeerJ since it may provide interesting information on vocal development in a fossorial mammal. Regarding the comments of three referees and myself your manuscript, however, needs substantial revision before being considered for publication.
Major objectives of all of us are related to the general framework, the technical soundness and the statistics used in your MS.

1. The background provided in the introduction and research questions addressed are not well linked to each other. E.g. You have not contributed to the role of learning in vocal development, neither can you link your data to the development of the vocal tract. Thus, provide a better framework for the research question you would like to address.
2. The Method part, in particular the description of the experimental design and set-up of your study as well as the acoustic recording and analysis system lacks more detailed information to make your study reproducible. Consult the current method section in the bioacoustics literature, e.g. Hammerschmidt et al. , Scheumann et al. for the acoustic recording and analysis part. Consult e.g. Scheumann et al. 2007, for the experimental design of social isolation experiments. This research is accessible via ResearchGate.
3. The results you presented are not always coming out from your study. See inconsistencies revealed in the comments of all of the referees.
4. The statistical approach needs major revisions, as pointed out by referee 1 and 3. Methods are not yet applied appropriately. Thus, consult the help of a statistician. With help of GLMMs it may be possible to explore the extent to which developmental changes in vocalizations are linked to body mass.

See detailed comments of the referees below. All the referees provided helpful suggestions on how improve your MS
In your rebuttal letter, please address carefully how you handle the comments of all three referees and myself.

Reviewer 1 ·

Basic reporting

The paper is generally well-written and organized. Multiple minor issues with word choice are listed with suggestions for improvement in General Comments. A couple of broader citations would improve scope of first paragraph of intro (see General Comments). The figures are very nice.

Experimental design

The study is of interest and, despite small sample sizes, definitely worthy of publication. The research questions are well-defined although the authors should consider whether their data can really address their second aim (vocal learning vs. vocal tract development). Additional information is needed in Methods (see General Comments).

Validity of the findings

Overall, the findings are valid because this is primarily a descriptive study. However, the statistical tests used are not appropriate to the data (see General Comments). The Discussion is quite good but I strongly encourage the authors to consider and discuss how vocal function and the context in which vocalizations are produced change over ontogeny in their own study species and others.

Additional comments

Title and throughout: The Anillaco tuco-tuco referred to as Ctenomys sp. Is this an undescribed species? If so, how is it distinguished from congeners? Some explanation that goes beyond referring readers to another publication from this research group is needed.

Abstract, L304: change undistinguishable to indistinguishable

Introduction
L46: “vocalizations experiment an ontogeny”. Experience? A better phrase might be, “vocalizations follow a developmental trajectory”
L47-51: Both citations here are for empirical papers. It would be better to have other citations to support these general opening statements.
L62: change “to” to “into”
L85: suggest removing “simply”
L88: change “can help understand how” to “can improve understanding of how”
L91: change “of” to “to”
L94: change “composed by” to “composed of”
L97: Not sure that abnormal is the right word. Abnormal has negative connotations for functionality whereas it sounds like these alternate vocal patterns are simply atypical/uncommon.

Materials and Methods
Please explain how juveniles were acoustically isolated from adults and address whether or not they were acoustically isolated from each other. Also, please provide some life history info that will place the term “juvenile” in a species-specific context: typical age at weaning (14 days seems very young), and age at sexual maturity for male and female Anillaco tuco-tucos. (Having read discussion I see that the latter is unknown; that should be stated here and at least a rough estimate could be provided based on other Ctenomys) Please provide sufficient information on the enclosure that readers can assess how the animals were maintained (e.g., group housed? Size of enclosure? Inside or outside?) and how acoustic data were collected without reading Amaya et al. 2016. This is particularly important for the “Recording of vocalizations” section. Were the animals recorded in their home enclosures or elsewhere? It sounds like they were recorded singly but this isn’t completely clear.

L162-170: Please clarify what 90% means in relation to duration and bandwidth.
L158-186: As I understand it, the way acoustic parameters were measured for triads and individual notes was very similar if not identical. Suggest combining the two sections to reduce redundancy in text.
L188-187: Some of this is redundant with lines 148-153.
L199: As above, consider changing abnormal to atypical.
L215: The Spearman correlation is not appropriate for the data because repeated measures collected from the same individuals are non-independent. It’s fine to plot the data as the authors have in Fig. 5 but to test for an effect of change in age (weight) on change in acoustic variables would need repeated measures ANOVA or a GLM with individual ID as a random variable. Same goes for the Kruskal-Wallis test.
L231-237: Ethics statement would be more appropriate at the end of Experimental animals and info on the stats package should go at the end of Statistical analyses.

Results
L243-244, 251 and elsewhere: change “were given” to “were produced” and “gave” to “produced”
L266: change “lumped” to grouped or split.
L271-272: This sentence is quite vague. Please be more specific as to how vocalizations achieved mature features at different developmental stages.
L280: change “conformed by” to “comprised of”
L282: “obviously” is subjective. A statistical test would be nice but if sample sizes make that impossible at least use numerical evidence. e.g., 7.8% is more than an order of magnitude > than 0.28% or, “abnormal” series were >25x more common in juveniles. Also, if reporting frequency of an event as a percentage should change proportion to percentage.

Discussion
L304-306: Need to acknowledge that, while this study did not find evidence for vocal learning, it also did not effectively test this hypothesis since the study animals were exposed to adult vocalizations for at least the first two weeks of life and it’s currently not clear whether juveniles were isolated from each other. This is addressed later in the Discussion, but if the authors want to include a statement about vocal learning in the first paragraph it should be modified.
L307: Again, how do the authors know that these “abnormal” vocalizations reflect poor motor control etc? How do they know that they don’t have a function in juveniles and/or adults? Without information on the context in which different types of vocalizations are produced, can’t assume that these vocal types are “mistakes”.
L332; 333-336: change “substantial genetic makeup in” to “substantial genetic contribution to”. Please clarify why the presence of series patterns in congeners supports the suggestion that there is a genetic basis to vocal patterns in the study species.
L337: tuning
L337-347: Again, I encourage the authors to consider how the function of vocalizations in their study species, and in other rodents, changes between birth and maturity.
L361: song has not been used previously to describe this species’ vocalizations, suggest changing.
L363: please clarify what “conformed only by series” means.
L369: Not all researchers agree that vocal learning is completely absent in rodents. See Jarvis 2006, Ornithological Science; Arriaga et al. 2012 PLoS ONE; Arriaga and Jarvis 2012 Brain and Language and papers citing these papers for ongoing debate.
L380: Listen is subjective. If the auditory meatus is open from birth then these animals can probably hear from birth. Standard ways to test for a contribution of learning to the development of adult vocalizations are complete acoustic isolation (very hard to do in mammals without deafening) and cross-fostering to a closely related species with different adult vocal patterns.
L391, 393: Change presumably to presumptively; “alleged” has negative connotations re. the conclusions of Camin 2010. Consider changing.
L397, 401: Georychus capensis
L403: occurs
L409: superficial?
L411, 413: Hooper, not Hopper. Also, this citation is missing from Bibliography.

·

Basic reporting

The manuscript is of interest, at elast for scholars and students interested in acoustic communications and rodents. It is sound, scientifically valid, well structured and celarly written. In my view, it fulfills all the criteria put on papers published in Peer J. I have only minor comments, although some of them (esp. that one concerning the age of the studied animals) must be addressed:
1) The authors state (Lines 122-130) that the captured juveniles were weaned individuals between 2-4 weeks old, and had weight of 33 g. According to Camin (2010), a paper the authors are also citing, weaning (in C. mendocinus) occus at the age of 56 d.a.b. and weight of 117 g. This discrepancy should be clarified.
2) (Lines 385-390) - It is argued that slow development in Ctenomys is due to its subterranean way of life. This is only partly true. More importantly - this is a common feature of all hystricognaths rodents!
3) Bathyergus siullius and Georinchus (line 397 and further on) - correct spelling is suillus and Georychus
4) unify writting vs - either with dot vs. or without vs
5) unify writing number and percent - either with space xx % or without xx%
6) In many cases there should be "the" after all: all the experiments, all the individuals, all the parameters etc. instead of all experiments, all individuals etc.
7) Line 46: I do not understand the sentence:
Like many other aspects of biological systems, vocalizations experiment an ontogeny.
(perhaps experience instead of experiment?)

Experimental design

I have no comments

Validity of the findings

The findings are valid

Additional comments

See above

Reviewer 3 ·

Basic reporting

see below

Experimental design

see below

Validity of the findings

see below

Additional comments

In general the manuscript present nice data showing developmental changes in the structure of long range vocalization of Anillaco Tuco Tuco. The problem with manuscript is the theoretical framework in which the author put their findings. Beside this there are some serious statistical problems.
1) Vocal learning vs. vocal maturation: The authors found no evidence for vocal learning which is not surprising because until now none of the studies describing developmental changes in rodent vocalizations found evidence of vocal learning. Their conclusion that the age related structural changes are related to maturational changes of the vocal tract is not very conclusive: First, the authors did not measure any vocal features which could describe changes related to vocal tract changes. Bandwidth and peak frequency do not describe formant structure. Insofar the authors have no measure whether changes in the vocal tract really lead to changes in the acoustic structure of these vocalizations. Further, looking at Fig. 5 makes clear that already juveniles at a weight of app. 80g have the same note structure as adults. The only difference which remains is sequence composition and number of notes. But this has nothing to do with vocal tract changes.
In addition it remains unclear to which degree adults and juveniles differ in sequence composition and number of notes. The tests (L266-270) seem to be wrong. In table S3 they make correct tests, one test per subject. But in this paragraph it seems that they combine the repeated measurements of their subjects. This cannot be tested with such tests. A chance level below 0.0001 having only three females is also a good sign that something is wrong. It is possible to use a simple pairwise test comparing old juveniles and adults.
2) Abnormal series in juveniles and adults: Independent whether it makes sense to use the term abnormal for the differences in the series, the major shortcoming is that the author forgot to take the motivation of the animals into account. The possibility that juveniles are already able to produce adult like vocalization but did not do it because they have no motivation is a general problem discussing factors underlying developmental changes. It is possible that they are already able to produce adult-like series but have no motivation.
3) Development of individual notes: The author found that only three subjects produced individual notes. None of the females produce single notes although they produce otherwise same notes as males. To me the meaning of these individual notes is totally unclear. Why are they produced only by these three males? Before making any developmental story out of it (“mixed developmental mode”), the author should try to find out whether there is any communicative relation with the production of these individual notes.
4) Life-history traits and territorial vocalizations …: This paragraph is very vague. In addition I cannot see why the development of the LRV is protracted about 110-120 days. Fig. 5 shows nicely that only very young subject differ in note structure. Juvenile seem to vocalize less but this could be simple due the fact that they have no motivation. If the authors could not come up with evidence that only the adults can produce the adult like version they should skip this part of the discussion.
5) It is unclear how the authors correct for multiple testing (e.g. Table S3).

---

## Round 0.2 · Minor Revisions

Please address the comments regarding statistics of ref 2 carefully and consult a statistician at your university to help you, in case it is not possible for you to do it by yourself. Without clarification the MS can not be accepted.

Reviewer 1 ·

Basic reporting

No comment

Experimental design

No comment

Validity of the findings

No comment

Additional comments

The authors have done a very nice and extremely thorough job of revising this manuscript. It is significantly improved and will make a nice addition to the literature on vocal ontogeny in rodents.

Reviewer 3 ·

Basic reporting

Most parts of the revised manuscript are clear and unambiguous.
The description of the different acoustic characterizations (L 199-239) is partly redundant, otherwise difficult to understand.
Similar table 2: Unclear why there are no p-values for “series per vocalization. What is the meaning of “A”, “B”, “C” & “D”?

Experimental design

no comment

Validity of the findings

The authors modified their statistical methods. Unfortunately not all changes led to a satisfactory result:
1) The authors present only p-values (table2, Text L304-313, 333-336). This is not correct. At least they should give also test statistic and/ or total N.
2) Again, it remains unclear whether the authors adjusted for multiple testing (pairwise statistic, several acoustic parameters).
3) It is unclear how it is possible to get a p-value below 0.001 with two subjects (females) using Friedman test.

---

## Round 0.3 · Minor Revisions

1. Introduction: Aim. use "may explain" instead of explain.
2. English language has to be improved. Please consult a native English speaker who will improve the English style of the MS and revises the minor errors.
3. Acoustic characterization. This new chapter helps in understanding what was done. Please give number of vocalizations you analyzed.
4. Your statistics does need improvement. Please consult a statistician. e.g. It is not allowed to perform a Wilcoxon test with N=2. To use this test to compare values between juveniles and adults with N=3 is also forbidden.
5. Make sure that the values from the adults are inside of the MS or in the Appendix. I do not see them.
6. Fig. 1, 2 3 scale on y-axis not visible. Make sure it is there.
7. Fig. 3 time scale on x-axis missing. Explain in legend what numbers in the figure mean.
8. Fig. 4 Give scale for x-axis
9. Fig. 5 Give scale for y-axis and for time and frequency on x-axis
10. No italics visible in Table 1. Make sure it is displayed
11. No adults displayed in Table 2. Include the respective values

---

## Round 0.4 · accepted · Accept

There are still problems with the English in your manuscript, but I hope the production staff can help. Congratulations on the first article.

Best wishes, Elke Zimmermanni